# Acoustical Properties of Fiberglass Blankets Impregnated with Silica Aerogel

**Hasina Begum** *[iD] and **Kirill V. Horoshenkov**

Department of Mechanical Engineering, The University of Sheffield, Sheffield S1 3JD, UK;
k.horoshenkov@sheffield.ac.uk
* Correspondence: hbegum3@sheffield.ac.uk; Tel.: +44-75-2157-0011

**Abstract:** It is known that aerogel impregnated fibrous blankets offer high acoustic absorption and thermal insulation performance. These materials are becoming very popular in various industrial and building applications. Although the reasons for the high thermal insulation performance of these materials are well understood, it is still largely unclear what controls their acoustic performance. Additionally, only a small number of publications to date report on the acoustical properties of fibrous blankets impregnated with powder aerogels. There is a lack of studies that attempt to explain the measured absorption properties with a valid mathematical model. This paper contributes to this knowledge gap through a simulation that predicts the measured complex acoustic reflection coefficient of aerogel blankets with different filling ratios. It is shown that the acoustic performance of a fibrous blanket impregnated with aerogel is generally controlled by the effective pore size and porosity of the composite structure. It is shown that there is a need for refinement of a classical Biot-type model to take into account the sorption and pressure diffusion effects, which become important with the increased filling ratio.

**Keywords:** acoustics; aerogels; modeling; fiber; porous materials

## 1. Introduction

There is a global need to reduce the use of fossil fuels and the release of greenhouse gases. Currently, 40% of energy consumption in Europe comes solely from the building sector [1], which is a major source of greenhouse gases. Due to this high level of energy consumption, the European council has introduced a 27% energy efficiency target for 2030 [2]. This has led to industries sourcing better energy-saving products for the market, with thermal insulation being the most effective way to reduce the energy consumption and loss. Achieving such a significant energy efficacy requires the development and upscaling of new commercial products based on aerogels.

One popular emerging thermal insulation product is aerogel blankets. Aerogel blankets consist of a silica aerogel embedded in a reinforcing fibrous matrix, which allows the brittle aerogel to become a flexible, durable solid used for buildings [3] and pipelines. The silica aerogel can undergo a surface modification process (typically hydrophobization) to enhance surface life stability [4], thus reducing the aerogel's susceptibility to moisture and rapid spoilage [5]. Silica aerogels themselves have porosity values as high as 98%, densities as low as 0.05–0.5 g/cm$^3$, surface areas in the range of 300–1000 m$^2$/g [6] and thermal conductivity values as low as 0.02 W/mK [7]. Application of aerogels on their own are limited due to their fragility and low mechanical modulus. Using them as composites in the form of aerogel blankets removes their fragility, as the aerogel grains are now incorporated within a fibrous matrix such as fiberglass or rockwool, giving them impeccable mechanical strength and a breadth of flexibility in terms of product development [8].

Conventional porous materials, such as nonwovens and polymer foams [9], can also prevent the reflection of sound incident waves to provide a high sound absorption performance [6,10] that is a highly desired property. Nonwovens in particular are ideal for

sound absorption due to their large surface area and high porosity, which offers increased frictional losses between sound waves and the fibrous matrix, leading to their good sound absorption performance [11].

Monolithic silica aerogels alone have unusual viscoelastic properties and have been used in the form of clamped plates to become the main source of intrinsic losses allowing them to exhibit subwavelength resonances for high sound absorption [12,13]. However, these materials are highly fragile. Utilizing them as aerogel powder into a fibrous, flexible matrix results in a multi-functional system that can fulfill a range of practical needs in many industry and domestic applications. Their fused nanoparticles in particular result in extremely low elastic stiffness, which provides a relatively low acoustic impedance and exceptionally low flexural wave speed, making it ideal for use as a subwavelength flexural element for controlling airborne sound [14]. Super-insulative acoustic absorbing materials such as aerogel blankets can be tailored and combined with other products to widen their applications and to provide lighter, thinner and more economical products.

It is known that the acoustic properties of aerogels alone are greatly influenced by the interstitial gas type, pore structure and aerogel density [15,16], and more recently the pioneering efforts to embed granular aerogels into a reinforced fibrous network [3] have shown promising acoustical behavior. The combination of the density and granular size of aerogel [17] and fiber reinforcement and decreased pore size greatly influences the sound absorption [18]. Motahari et al. [19] investigated the aging time of silica aerogels in cotton nonwoven mats on the sound absorption performance. They found that the presence of low density (0.088 g/cm$^3$) silica aerogel at different molar ratios of the precursors MeOH/TEOS used and the low aging time enhanced the sound absorption coefficient in the low frequency range of 250 to 2500 Hz. Furthermore, Eskandari et al. [20] investigated the acoustical behavior of synthesized silica aerogels mixed into UPVC blankets of different weight ratios. They found that neat UPVC only had a maximum sound absorption of 17% at a frequency of 1800 Hz; however, when silica aerogel was applied at 0.5, 1.5 and 3 weight %, the maximum sound absorption of UPVC increased to 24, 28 and 43%, respectively, therefore highlighting that acoustical properties were greatly increased upon the addition of silica aerogel. A more extensive review of acoustical properties of aerogels can be found in reference [4].

However, there is a general lack of understanding regarding what leads to the observed acoustical properties of granular aerogels embedded into fibrous mats. A majority of previous works have not attempted to apply any valid theoretical models to predict key acoustical properties of these systems to explain the measured data. There are limited data on the effect of the filling ratio on the acoustical properties of aerogel impregnated fibrous blankets. Additionally, despite some previous efforts (e.g., [6,9]), there is a limited understanding on the ability of some prediction models to explain the general acoustical behavior of these materials. There was no discussion on the values of the non-acoustical parameters that the authors of references [6,9] had to use in the prediction models they chose in their works to simulate the measured absorption coefficient data.

Our work aims to address this gap via a careful characterization of the acoustical behavior of granular silica aerogels impregnated into fiberglass mats. The acoustical properties of five samples of aerogel blankets with varying concentrations of aerogel powder (at micrometric particle sizes) at filling ratios of 0, 25, 50, 75 and 100% were measured and predicted using a mathematical model. This work helps to better understand the relation between their micro-structure and measured acoustical performance.

The structure of this paper is as follows. Section 2 highlights the various techniques used to characterize the chemical and physical material properties of aerogel blankets. Section 3 looks at the experimental acoustical data derived from the analysis of these materials. Section 4 attempts to explain these data with a mathematical model to understand what intrinsic properties of aerogel blankets make them acoustically absorbing.

## 2. Materials and Methods

### 2.1. Materials Preparation

As specified in the patent [21], sodium silicate diluent was prepared using distilled water to achieve 3 to 10 weight % of $SiO_2$ and stirred with hexamethydisilazane (HDMS) whilst slowly adding nitric acid ($HNO_3$) to allow gelation to occur. The silylated hydrogel and co-precursor were gradually immersed in n-hexane for a one-step solvent exchange and sodium ion removal. Water present in the hydrogels is detached due to surface modification of the organic groups ($-CH_3$)$_3$ in HDMS. The hydrogel from which water was removed was then dried at ambient pressure and pulverized to form a superhydrophobic synthetic silica aerogel powder [22] with the particle diameter in the range of 1–20 μm impregnated into a fiberglass blanket at different weight % of 25, 50, 75 and 100 powder to blanket at a later manufacturing process. This is a standard process [23]. The fiber diameter in the blanket was 10 microns and its density was approximately 73 kg/m$^3$. It is a standard commercial E-glass fiber needle mat produced by Lih Feng Jiing Enterprise Co Ltd (Tainan City, Taiwan) [23]. The impregnated fiberglass blankets were then cut to a 10 mm diameter size using a hand-held hole saw that had smooth blade edges to ensure a perfect fit into the impedance tube when tested for acoustical properties.

### 2.2. Materials Characterization

Microstructural observations such as particle distribution of the silica aerogels within fiberglass mats were performed using scanning electron microscopy (SEM). Images were obtained with a FEI Nova NanoSEM 230 instrument (FEI, Hillsboro, OR, USA) at an accelerating voltage of 10 kV and a minimum working distance of 5 mm. The silica aerogels were fixed on the sample holder using a carbon pad and subsequently coated with 15–20 nm of platinum for SEM analysis.

The acoustical properties of aerogel blankets were measured in a 10 mm impedance tube that was custom made by Materiacustica [24]. This 2-microphone tube setup was developed to test small material specimens in accordance with the standard ISO 10534-2:2001 [25]. This setup enabled us to measure the normalized surface acoustic impedance, complex reflection coefficient and sound absorption coefficient of a hard-backed porous layer in the frequency range of 300–3000 Hz. The spacing between the two microphones was 30 mm, which is usual for this frequency range as recommended in the standard [25]. The thickness of the samples used in the acoustic experiments was between 7 and 11 mm, which is a typical thickness of a commercial product [23,26]. Figure 1 illustrates a typical specimen of fiberglass blanket impregnated with aerogel that was used in the acoustic experiments. Figure 2 shows a photograph and jigsaw drawing of the vertically standing impedance tube.

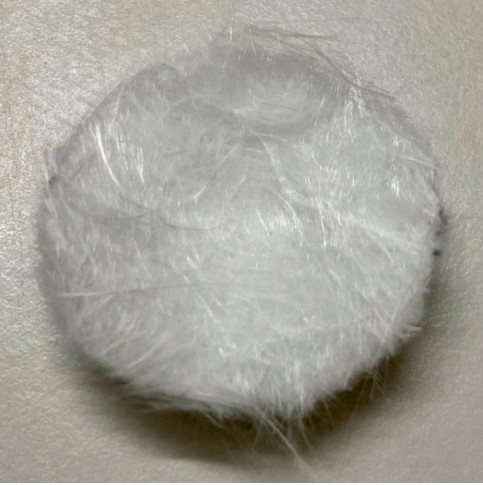

**Figure 1.** 10 mm diameter of fiberglass blanket samples cut for fitting into the impedance tube.

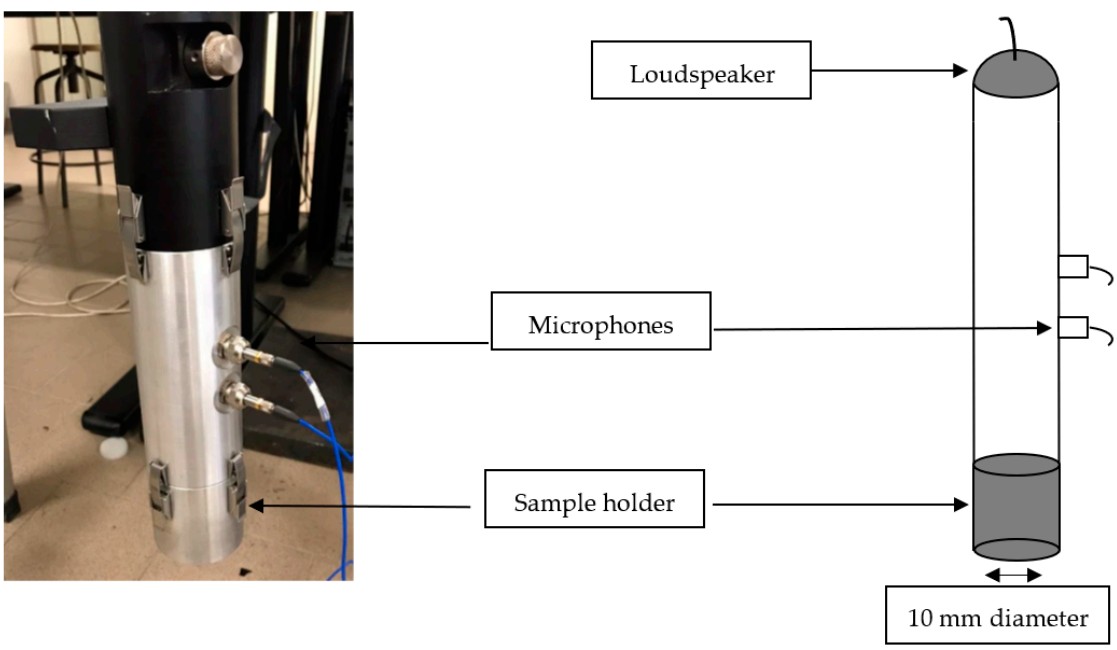

**Figure 2.** 2-Microphone impedance tube setup to measure the surface impedance of a porous layer [25].

## 3. Modeling of the Acoustical Properties of Fibrous and Granular Media

Basic modeling of the acoustical properties of this kind of material requires a mathematical model that takes into account the classical visco-thermal effects in the voids' between the fibers and loose granules of powder. However, a fibrous blanket impregnated with aerogel is a more complicated void structure that has at least three scales of porosity. The fiberglass blanket itself consist of 10 μm interlaced fibers that form a porous structure with sub-millimeter size pores of approximately 0.1 mm. The aerogel particles are around 20 μm in size and contain nano-pores of 20 nm in size.

There are several models that exist that can predict the acoustical properties of classical fibrous media [27]. In this work we attempt to use the model proposed by Horoshenkov et al. [28], which is based on the following three parameters: (i) the median pore size, $\bar{s}$; (ii) porosity, $\phi$; and (iii) the standard deviation in pore size, $\sigma_s$. This reduced number of parameters allows easier inversion of key morphological characteristics of porous media from acoustical data. This model predicts the dynamic density, $\tilde{\rho}$, and complex compressibility, $\tilde{C}$, of air in the material pores. These quantities are given by the analytical equations, which are presented in reference [28]. The MATLAB code to predict these quantities can be found in reference [29].

The normalized surface impedance of a hard-backed layer of porous material that is typically measured in the impedance tube is:

$$Z_s = -jZ_c \cot(k_c d)/\rho_0 c_0 \tag{1}$$

where $j = \sqrt{-1}$, $d$ is the sample thickness, $\rho_0$ is the ambient density of air, $c_0$ is the sound speed in air,

$$Z_c = \sqrt{\frac{\tilde{\rho}}{\tilde{C}}} \tag{2}$$

is the characteristic impedance and

$$k_c = \omega\sqrt{\tilde{\rho}\tilde{C}} \tag{3}$$

is the wavenumber in the porous material. Here, $\omega$ is the angular frequency of sound. In this work, we use the complex reflection coefficient data

$$R = \frac{Z_s - 1}{Z_s + 1} \tag{4}$$

to fit the model. The work presented in reference [30] shows that the complex reflection coefficient is a reliable quantity to determine the effective values of the three non-acoustical parameters in the model [28] through the parameter inversion. This is the complex acoustical quantity that is measured directly using the standard impedance tube method [25]. The real and imaginary parts of this quantity are bounded between $-1$ and $+1$, which makes them attractive to use in the parameter inversion process. The complex reflection coefficient can also be used to predict the acoustic absorption coefficient

$$\alpha = 1 - |R|^2 \tag{5}$$

which is a usual measure of the ability of the porous layer to absorb sound.

## 4. Results and Discussion

### 4.1. Microstructural Analysis

Figures 3–7 present SEM images of the fiberglass blankets with a progressive increase in the aerogel powder filing ratio from 0 to 100%. These images can be used to identify the aerogel particle distribution in fiberglass blankets and the structure of the fiber network. The SEM magnification scale in each of these images changes between 40, 100 and 500 microns to provide a better view inside into the microstructure. We note that SEM image analysis is sensitive to the loading of samples on to the carbon stub; a large amount deposited will affect the coating and this may fracture the image surfaces. Furthermore, there may be sampling bias causing the contrast/brightness settings to be adjusted and this may also affect the results.

Figure 3 clearly shows that there is little to no aerogel powder present in virgin fiberglass. It also shows that the spacing between individual fibers is in the order of 100 s of microns and that these randomly oriented fibers form a complicated network. The addition of a relatively small (25%) amount of aerogel powder does not significantly affect the inter-fiber space (see Figure 4). For this case, aerogel particles mainly attach themselves to the fibers (see Figure 4a) causing an apparent increase in the fiber diameter (see Figure 4). In the case of the samples with 50 and 75% concentrations (Figures 5 and 6, respectively), a similar effect can be visually observed, but the apparent increase in the fiber diameter is more significant whereas the size of the inter-fibrous space is clearly reduced. In the ultimate case, when the aerogel filling ratio in the fibrous sample is 100% (see Figure 7), a considerable proportion of the inter-fibrous space is occupied with aerogel powder so that the effective pore size appears to be significantly reduced visually.

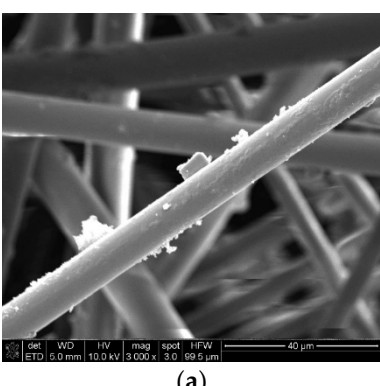 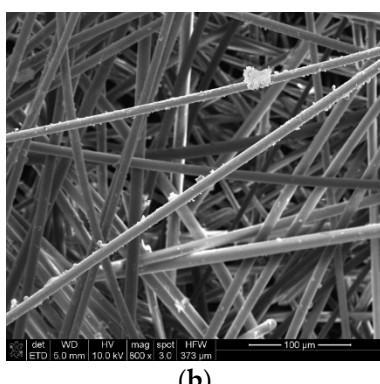 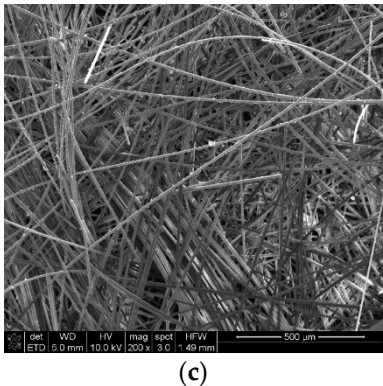

     (**a**)              (**b**)              (**c**)

**Figure 3.** SEM images taken at different magnifications ($3000\times$ (**a**), $800\times$ (**b**) and $200\times$ (**c**)) showing the fiberglass blanket without any aerogel.

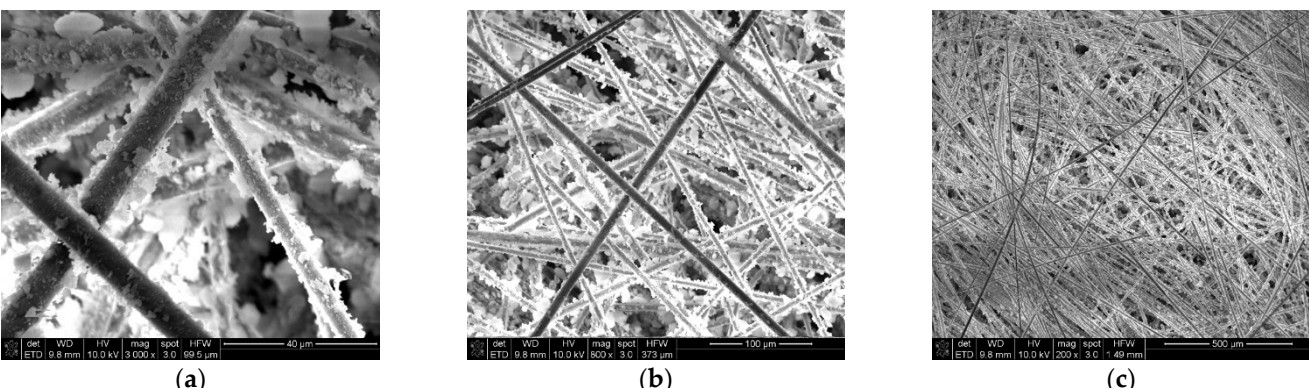

(**a**)             (**b**)             (**c**)

**Figure 4.** SEM images taken at different magnifications (3000× (**a**), 800× (**b**) and 200× (**c**)) showing the fiberglass blanket structure with an aerogel filling ratio of 25%.

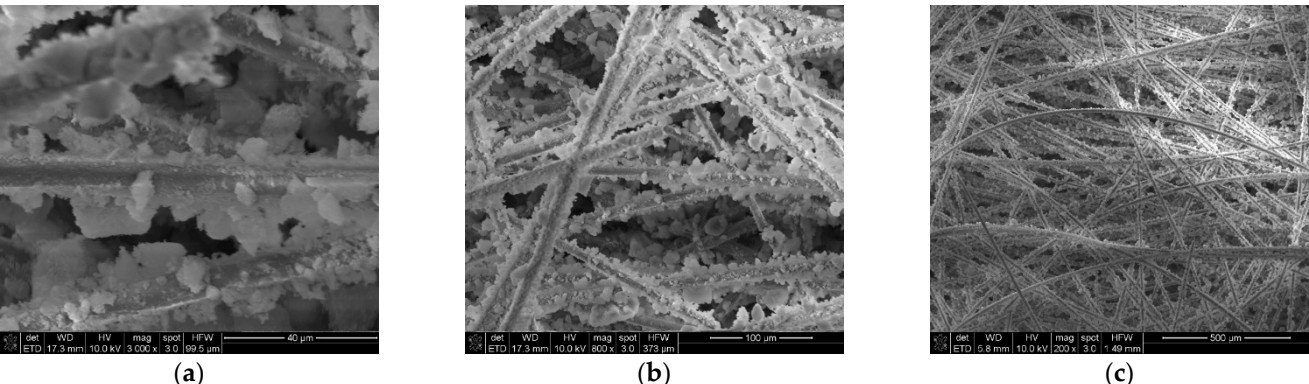

(**a**)             (**b**)             (**c**)

**Figure 5.** SEM images taken at different magnifications (3000× (**a**), 800× (**b**) and 200× (**c**)) showing the fiberglass blanket structure with an aerogel filling ratio of 50%.

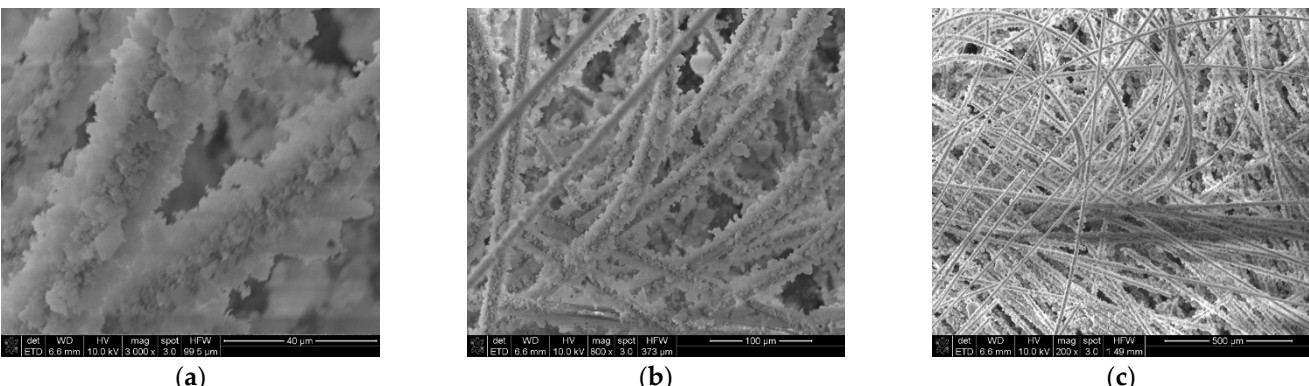

(**a**)             (**b**)             (**c**)

**Figure 6.** SEM images taken at different magnifications (3000× (**a**), 800× (**b**) and 200× (**c**)) showing the fiberglass blanket structure with an aerogel filling ratio of 75%.

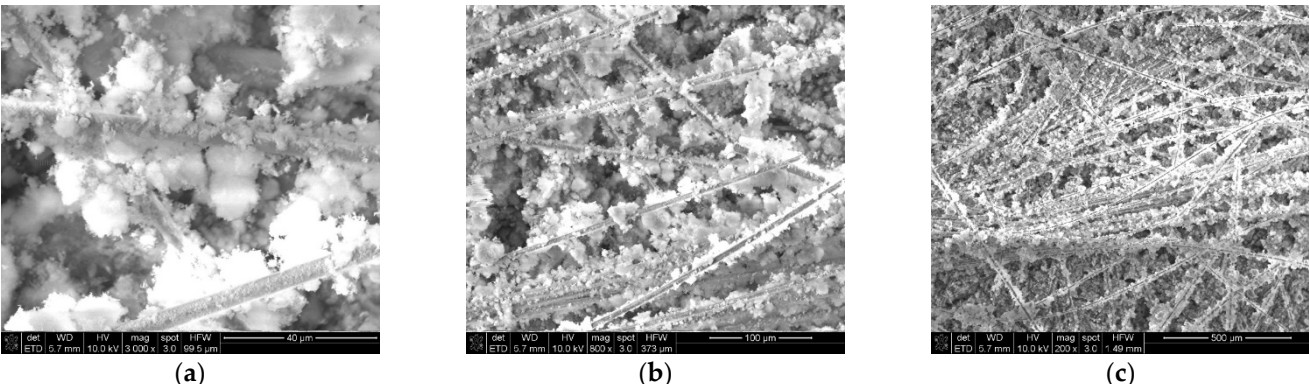

**Figure 7.** SEM images taken at different magnifications (3000× (**a**), 800× (**b**) and 200× (**c**)) showing the fiberglass blanket structure with an aerogel filling ratio of 100%.

### 4.2. Acoustical Properties

The acoustical properties were measured at the University of Sheffield in a 10 mm impedance tube [25]. Five specimens were cut from different areas on a sample of each type of fibrous blanket and their properties were measured. The repeatability of each measurement was found within ±2.9% for the absorption coefficient and ±5.8% for the reflection coefficient. Figure 8 shows a comparison between the measured absorption coefficients for the five samples. Figure 9 presents a comparison between the measured and predicted real and imaginary parts of the complex reflection coefficients for these five materials.

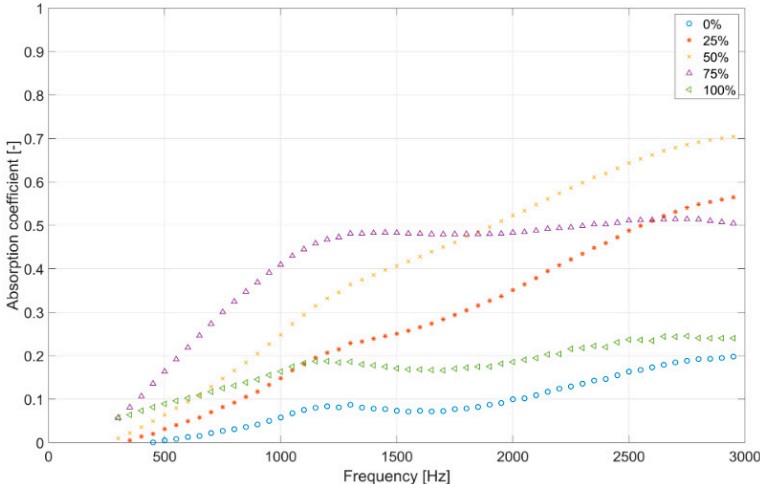

**Figure 8.** An example of the measured sound absorption coefficient of a 8–9 mm thick hard-backed layer of the five fibrous blankets with a progressive increase in the aerogel filling ratio.

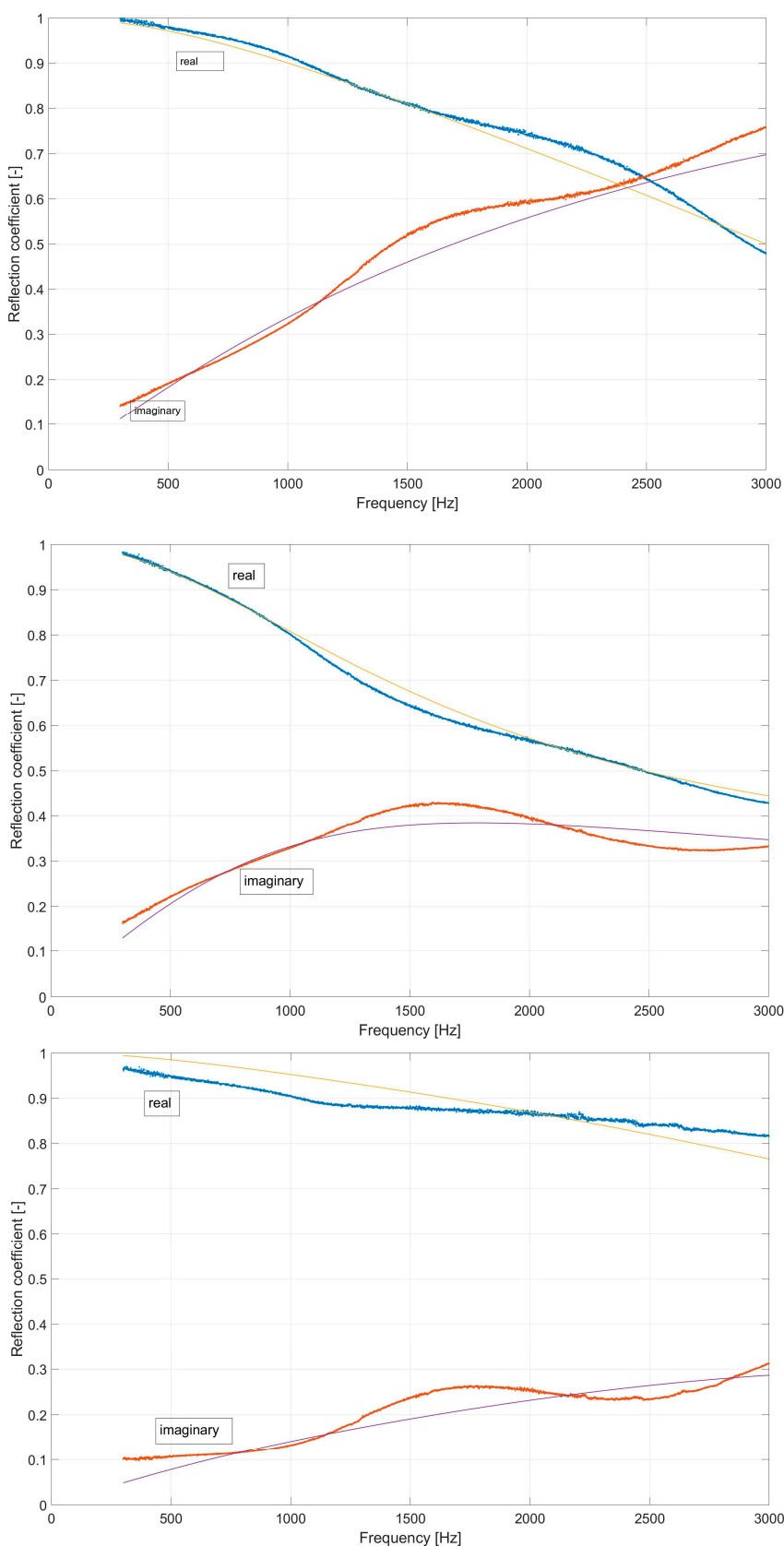

**Figure 9.** Examples of the measured (marker) and predicted (solid lines) complex reflection coefficient data for fiberglass blanket without any aerogel (**top**), 50% aerogel impregnated blanket (**middle**) and 100% aerogel impregnated blanket (**bottom**).

The results presented in Figure 8 suggest that there is a progressive increase in the absorption coefficient as the aerogel impregnation increases from 0 to 50%. When the aerogel filling ratio reaches 75% this increase becomes less pronounced. Increasing the filling ratio beyond 75% reduces the absorption coefficient significantly. This reduction makes sense because it is likely associated with a densely packed inter-fibrous space, which becomes almost full with aerogel (see Figures 6 and 7), causing a considerable reduction in the pore size (see Table 1) in a relatively thin material layer. For the filling ratios of 75% and above the characteristic impedance (see Figure 9) and attenuation of sound in a layer with such small pores becomes very high, limiting the value of the absorption coefficient [27]. The absorption coefficient of this relatively thin fibrous blanket with 50–75% filling ratios is still relatively high (30–70%) particularly above 1000 Hz (see references [4,13]). This level of absorption has a practical value in applications related to engineering noise control.

**Table 1.** Values of the non-acoustical parameters inverted from fitting the model [28] to the measured complex reflection coefficient data for the five types of fiberglass blankets.

| Filling Ratio, % | Layer Thickness, $d$, mm | Pore Size, $\bar{s}^{(i)}$, μm | Porosity, $\phi^{(i)}$ | Standard Deviation in Pore Size, $\sigma_s^{(i)}$ | Calculated Porosity, $\phi$ | RMS Error, % |
|---|---|---|---|---|---|---|
| 0 | 8.12 ± 0.77 | 99.4 ± 4.15 | 0.994 ± 0.0098 | 0 | 0.965 ± 0.0041 | 1.4 |
| 25 | 9.33 ± 1.60 | 48.0 ± 20.2 | 0.938 ± 0.018 | 0.160 ± 0.213 | 0.960 ± 0.0044 | 1.7 |
| 50 | 9.26 ± 0.47 | 32.8 ± 2.00 | 0.929 ± 0.011 | 0 | 0.952 ± 0.0026 | 1.8 |
| 75 | 10.35 ± 0.85 | 20.5 ± 1.35 | 0.959 ± 0.032 | 0 | 0.951 ± 0.0036 | 3.6 |
| 100 | 9.34 ± 0.84 | 83.0 ± 2.04 | 0.505 ± 0.091 | 0.55 ± 0.015 | 0.94 ± 0.0067 | 2.5 |

An obvious question here is: *What happens to the fiberglass pore properties when the percentage of aerogel powder impregnating the blanket increases?* In order to answer this question we attempted to fit the mathematical model [28] to the complex acoustic reflection coefficient data measured in the impedance tube. We used the optimization procedure described in reference [30] to invert the three parameters of the best fit. This procedure has been used extensively by many researchers (see [27] for a review of parameter inversion methods). Figure 9 shows three examples of this fit for fiberglass blankets with aerogel filling ratios of 0, 50 and 100%.

Table 1 presents a summary of the mean values of the three non-acoustical parameters in the adopted theoretical model [28], which were inverted from its fit to the measured data for the five filling ratios. This table also provides the porosity values calculated from the material density data, mean layer thickness measured directly and root mean square error (RMS) calculated between the predicted and measured reflection coefficient spectra. The superscript $^{(i)}$, which appears with a non-acoustic parameter in this table, means that the values of this parameter were inverted rather than measured directly.

The results shown in Figure 9 and the parameter values listed in Table 1 suggest that the model generally provides a very close fit to the data (an RMS error better than 2.5%), particularly when the filling ratio is equal to or below 50%. The agreement between the predicted and measured reflection coefficient spectra reduces slightly with the increased filling ratio. The inverted value of the median pore size (Table 1) decreases progressively from 99.4 to 20.5 μm as the filling ratio increases from 0 to 75%. This makes physical sense, as the SEM images in Figures 3–7 illustrate this. This range of pore sizes is also consistent with that measured non-acoustically for similar materials [6]. When the filling ratio increases, the inter-fiber pores are progressively replaced with much smaller inter-grain pores. The transport (inner) pores in the grains of aerogel do not seem to contribute significantly to the measured acoustical properties. This is reflected in a consistently underpredicted porosity value, $\phi^{(i)}$. The progressive change in the inverted porosity value make sense for the filling ratios between 0 and 50%, dropping from 99.4 to 92.9%,

respectively. These values match the measured porosity values within 3%. When the filling ratio increases to 100%, the inverted porosity of $\phi^{(i)} = 50.5\%$ is significantly below the measured porosity of $\phi = 93.6\%$. Additionally, the median pore size inverted for this type of blanket is not realistic. This suggests that the physical behavior of the blanket layer with 100% filling ratio is no longer captured accurately by the model. As the proportion of aerogel powder in the material approaches 100%, the sorption and thermal diffusion effects are likely to become much more important [31]. These effects are not captured by the adopted model [28], which only accounts for the classical visco-thermal and inertia effects.

## 5. Conclusions

This work is a systematic study of the acoustical properties of fibrous blankets that are impregnated with an aerogel powder. The level of impregnation (filling ratio) has been progressively changed from 0 to 100% with respect to the material weight. The complex acoustic reflection coefficient of these materials was measured in the frequency range of 300–3000 Hz using a standard impedance tube setup [25]. These data were used to invert the three parameters of the theoretical model [28] via the best fit method [30]. It was found that the adopted model can predict the reflection coefficient spectrum relatively accurately with the RMS error being below 4%. The absorption coefficient of these relatively thin (8–9 mm thick) fibrous blankets with 50–75% filling ratios is relatively high (30–50%), particularly above 1000 Hz. This level of absorption has a practical value in applications related to engineering noise control.

The results of the parameter inversion obtained with the adopted model suggest that the impregnation of fibrous blanket with an aerogel powder results in a progressive reduction in the effective pore size. For the filling ratios in the range of 0–50% there is also a small but progressive reduction in the inverted porosity, which is within 3% of that measured directly. The absorption coefficient increases progressively with the increased filling ratio, reaching its maximum when the filling ratio is between 50% and 75%. This decrease in the effective pore size results in an increased acoustic attenuation and better coupling, which are important to maximize the acoustic absorption for such a thin porous layer. Increasing the filling ratio beyond 75% results in a significant drop in the absorption. This drop is associated with a considerable drop in the porosity value ($\phi = 0.505$) and substantial increase in the pore size ($\bar{s} = 83\ \mu m$) inverted for the filling ratio of 100%. The discrepancy between the model and data for this filling ratio increases. As the proportion of aerogel powder in the material approaches 100%, the open porosity does not drop significantly, i.e., the proportion of the open interconnected pores remains relatively constant. However, the sorption and thermal diffusion effects in the inner pores in the aerogel grains become much more important [31]. These pores have nanometer scales [6], which is much smaller than the values of $\bar{s}$ inverted with the model [28] (see Table 1). The effects that occur in nanometer pores cannot be captured by the adopted model [28], which only accounts for the classical visco-thermal and inertia effects in pores that are much larger than the mean free path (68 nm in air at ambient pressure and temperature).

This work suggests that in order to predict the acoustic behavior of fibrous blankets with high aerogel filling ratios there is a clear need to refine the model [28] to include the sorption and pressure diffusion effects. The adopted model does require unrealistic values of the median pore size and porosity to achieve a good fit. This model can be refined by including in it the work by Venegas and Umnova [31]. In this way the dynamic compressibilities of the air filling the inter-fiber pores and in the nanoscale pores in the aerogel grains can be combined to account for all of the physical effects that contribute to the observed acoustical behavior.

**Author Contributions:** Conceptualization, H.B. and K.V.H.; methodology, H.B.; software, K.V.H.; validation, formal analysis, H.B.; investigation, H.B.; resources, H.B.; data curation, H.B.; writing—original draft preparation, H.B.; writing—review and editing, K.V.H.; visualization, H.B.; supervision, K.V.H.; project administration, K.V.H.; funding acquisition, K.V.H. All authors have read and agreed to the published version of the manuscript.

**Funding:** This research was partly funded by the EPSRC-sponsored Centre for Doctoral Training in Polymers, Soft Matter and Colloids, grant number EP/L016281/1, and industry sponsors—Armacell.

**Institutional Review Board Statement:** Not applicable.

**Informed Consent Statement:** Not applicable.

**Data Availability Statement:** The data are available online (see ref. [29]).

**Acknowledgments:** The authors would like to thank the EPSRC-sponsored Centre for Doctoral Training in Polymers, Soft Matter and Colloids at The University of Sheffield for their financial support of this work. We would also like to thank our industry partner Armacell and Mark Swift and Pavel Holub for their continued support throughout this research study. We extend our thanks to Shanyu Zhao at the Swiss Federal Laboratories for Materials Science and Technology for allowing us to use their electron microscopy center for high magnification SEM image analysis.

**Conflicts of Interest:** The authors declare no conflict of interest. The funders had no role in the design of the study; in the collection, analyses, or interpretation of data; in the writing of the manuscript, or in the decision to publish the results.

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
