# Peer review of "Acoustical Properties of Fiberglass Blankets Impregnated with Silica Aerogel"

_applsci, doi:10.3390/app11104593_

Round 1

Reviewer 1 Report

The work is well done, clear, well written and complete with every necessary part. The figures are clear and well done. The methodology of study correct. The topics covered, although not very original, are interesting and for this reason I recommend the publication without any particular changes.
I suggest only to check at line 136 the unit of measure 100 mm that probably is not correct due to a typing error.

Reviewer 2 Report

The work describes the experimental characterization of fiberglass blankets impregnated with silica aerogel. The results are well presented and the article is clear and well written. I found very interesting the possibility of impregnating fiber or porous materials (open cell ones) with aerogel powders to increase their capabilities. The article has the standards of the journal, but it needs some minor corrections, shown as follows:

1.- The goal of the work is to discuss the acoustic properties of these materials. In my opinion, the introduction should discuss the use of aerogel in acoustics, in particular to increase the acoustic absorption. I suggest adding and commenting the following references:

- Phys. Rev. Applied, 5, 034012, (2016)

- Appl. Phys. Lett., 115, 061901, (2019)

- Journal of Non-Crystalline Solids 499, 283–288, (2018)

2.- In the first paragraph of section 2.1 authors say that the particle diameter is in the range of 1-20 mm. This seems too big to me.

Later, in section 3 authors claim that the fiberglass blanket has 10 µm fibers with aerogel particles of 20 nm.

Can the authors give more details or be clearer?

3.- Section 2.2. Figure 1 and text corresponding to Figure 1 talk about aerogel blanket, however I think this is the fiberglass blanket.

4.- Figure 8 shows the evolution between the different analyzed samples. It is shown that 75% shows a limit for absorption. Can the authors discuss what will be the real maximum?

5.- Figure 9. The size of the fonts in the axis, legends etc… is too small. Please increase it.

6.- A discussion on the case with 100% of filling fraction is needed. In this case, the system becomes a close pore material? Please provide some more details.

7.- Authors claim that for 50-75% filling ratios the absorption coefficient is relatively high particularly above 1000 Hz. What are the authors using as comparison? The results should be compared with existing materials.

Typos:

Line 149: “d is” Instead of “dis”.

Line 173. “Fig. 3 clearly shows” instead of “Figs. 3 clearly show”

Line 212. “Fig. 9 shows” instead of “Figs. 9 show”

Line 214. “Table 1 presents” instead of “Table 1 present”

Line 225. “The inverted value of the median pore size (Table 1) decreases” instead of “The inverted value of the median pore size (Table 1) decrease”

Reviewer 3 Report

This paper presents an analytical approach to evaluate the acoustic properties of fibrous blanket impregnated with aerogel. The paper is well written and the conclusions are supported by the presented results. I only have a few remarks which I believe may further improve the quality of the paper.

1) a few typos can be spotted throughout the manuscript, please double-check english grammar.

2) please double-check the units of aerogel particles and fibreglass diameters, in lines 102-105 and 135-137.

3) it might be beneficial for the reader if the authors could provide the analytical expressions of the three parameters of the adopted model.

4) The quantity \omega in equation 3) should be properly defined as angular frequency.

5) could the authors please provide a more detailed description of the algorithm used  to determine the non-acoustical parameters through the inverse procedure? 
